# An Artificial Intelligence-Enabled ECG Algorithm for Predicting the Risk of Recurrence in Patients with Paroxysmal Atrial Fibrillation after Catheter Ablation

**DOI:** 10.3390/jcm12051933

**Published:** 2023-03-01

**Authors:** Junrong Jiang, Hai Deng, Hongtao Liao, Xianhong Fang, Xianzhang Zhan, Wei Wei, Shulin Wu, Yumei Xue

**Affiliations:** Guangdong Cardiovascular Institute, Guangdong Provincial People’s Hospital, Guangdong Academy of Medical Sciences, Guangzhou 510080, China

**Keywords:** artificial intelligence, atrial fibrillation, catheter ablation, recurrence

## Abstract

**Background:** Catheter ablation (CA) is an important treatment strategy to reduce the burden and complications of atrial fibrillation (AF). This study aims to predict the risk of recurrence in patients with paroxysmal AF (pAF) after CA by an artificial intelligence (AI)-enabled electrocardiography (ECG) algorithm. **Methods and Results:** 1618 ≥ 18 years old patients with pAF who underwent CA in Guangdong Provincial People’s Hospital from 1 January 2012 to 31 May 2019 were enrolled in this study. All patients underwent pulmonary vein isolation (PVI) by experienced operators. Baseline clinical features were recorded in detail before the operation and standard follow-up (≥12 months) was conducted. The convolutional neural network (CNN) was trained and validated by 12-lead ECGs within 30 days before CA to predict the risk of recurrence. A receiver operating characteristic curve (ROC) was created for the testing and validation sets, and the predictive performance of AI-enabled ECG was assessed by the area under the curve (AUC). After training and internal validation, the AUC of the AI algorithm was 0.84 (95% CI: 0.78–0.89), with a sensitivity, specificity, accuracy, precision and balanced F Score (F1 score) of 72.3%, 95.0%, 92.0%, 69.1% and 0.707, respectively. Compared with current prognostic models (APPLE, BASE-AF2, CAAP-AF, DR-FLASH and MB-LATER), the performance of the AI algorithm was better (*p* < 0.01). **Conclusions:** The AI-enabled ECG algorithm seemed to be an effective method to predict the risk of recurrence in patients with pAF after CA. This is of great clinical significance in decision-making for personalized ablation strategies and postoperative treatment plans in patients with pAF.

## 1. Introduction

Atrial fibrillation (AF) is one of the most common arrhythmias in adults, with an estimated prevalence ranging from 2% to 4% [1]. The prevalence is expected to rise further due to increased longevity in the general population and further research into undiagnosed atrial fibrillation [2]. Catheter ablation (CA) is an important treatment strategy to reduce the burden and complications of AF. The recurrence after catheter ablation is the result of the interaction of many factors, including the duration of AF, age, left atrial (LA) size, renal insufficiency, atrial fibrosis, operators, etc. [3]. Current prognostic models have shown potential benefits in predicting the risk of recurrence in patients with AF after CA [4,5]. However, these models can only moderately predict recurrence, and no model has been proven to be superior to others. Therefore, it is of great clinical significance to establish an accurate and effective model with high clinical applicability and generalization to predict the risk of recurrence in patients with AF after CA.

With the development of deep learning models, an artificial intelligence (AI) algorithm is gradually applied to cardiovascular disease [6,7,8]. Some studies focusing on using machine learning to predict the risk of AF recurrence after CA showed that AF recurrence after CA can be predicted from clinical records [9,10,11], atrial shape changes [12] and the combination of AF simulation results and imaging features [13]. Electrocardiography (ECG) is an excellent substrate for deep learning models because ECG data with finite complexity is obtained in consistent protocols and archived in usable digital formats. The application of AI in ECG has become a non-invasive and low-cost method for the diagnosis of cardiovascular disease. We hypothesized that high-dimensional ECG features extracted by an AI algorithm might be able to reflect changes associated with potential recurrence. Therefore, we first tried to develop and validate an AI-enabled ECG algorithm to predict the risk of recurrence after CA by using preoperative 12-lead ECG.

## 2. Materials and Methods

### 2.1. Data Sources and Study Population

In this study, 2530 ≥ 18 years old patients with pAF who underwent CA in Guangdong Provincial People’s Hospital from 1 January 2012 to 31 May 2019 were retrospectively collected. These patients obtained at least one standard 10 s, 500 Hz, 12-lead sinus rhythm ECG within 30 days before CA. 1705 patients were followed up for more than 12 months. 87 patients were excluded due to the lack of original 12-lead ECG data. All patients underwent pulmonary vein isolation (PVI) by experienced operators. Baseline clinical features were recorded in detail before CA, and standard follow-up was conducted. 12-lead ECG and Holter were performed at 1, 2, 3, 6 and 12 months after CA. Echocardiography was performed at 3, 6 and 12 months after CA. Any ≥30 s atrial tachyarrhythmias that occurred within 3 months after CA were defined as early recurrence (ER) while those that occurred within 3–12 months after CA were defined as late recurrence (LR). The recurrence in this study referred to LR. For patients with multiple 12-lead ECGs obtained 30 days before CA, the 12-lead ECG closest to the operation date was selected. The study was approved by the ethics committee of Guangdong Provincial People’s Hospital and was conducted in accordance with the Declaration of Helsinki and Good Clinical Practice Guidelines. The requirement for informed consent from patients whose information was retrospectively collected was waived. In order to train and validate the convolutional neural network (CNN) model, the patients were randomly divided into the training set (n = 971), the validation set (n = 162) and the testing set (n = 485) at proportions of 6:1:3 (Figure 1). The training set was used to train the neural network, the validation set was used to optimize the network and select the parameters, and the testing set was used to evaluate the performance of the neural network. ECGs in different datasets were not repeated.

### 2.2. Model Development

Before training, the 12-lead ECG data were preprocessed to remove the baseline drift and noise [14]. Each ECG was a 12 × 5000 (12 leads by 10-s duration sampled at 500 Hz) matrix. In order to produce more data, the training and validation set was expanded by shifting the start point and choosing continuous 4500 points of each lead, which contained 5000 points in total, so it could be expanded to 500 samples for each sample by shifting the start point 500 times [15]. The same amounts of 12-lead ECGs with recurrence and without recurrence were randomly selected as the input batch. 12-lead ECGs in different datasets were not repeated. Convolutional neural networks (CNNs) were built by using the Keras Framework with a TensorFlow backend and Python. The categorical cross-entropy loss was used as the loss function, and the Adam optimization method was applied. Multiple networks were tested and the simplest (the one with fewer parameters or layers) that resulted in the highest AUC was selected. The model that provided the optimal AUC was similar in concept and structure to a previous model developed to identify left atrial enlargement with 12-lead ECG (Figure 2) [14]. The first six layers were convolution layers with 32, 32, 64, 64, 128 and 128 filters, respectively. The shapes of the filters were composed of 5 × 1 and 3 × 1, alternately. “Relu” activation, batch-normalization and max-pooling were used after each convolution layer. Then the last convolution layer was shaped as 12 × 1 to fuse data from the different leads. After convolution, the data were fed to a dropout layer and a fully connected network.

### 2.3. Statistical Analysis

The risk of recurrence was assessed by current scoring systems (APPLE [16,17,18], BASE-AF2 [19], CAAP-AF [20], DR-FLASH [21], MB-LATER [22,23,24]) (Appendix A). Compared with different scoring systems and clinical prognostic factors, the predictive performance of the AI algorithm was evaluated. The continuous variables were compared by using the independent t-test, and the 2-group categorical variables were compared by using the χ2 test and the log-rank test. Independent prognostic factors were determined by stepwise logistic regression analysis. Measures of performance included the area under the curve(AUC), accuracy, precision, sensitivity, specificity, and a balanced F Score (F1 score). The receiver operating characteristic curve (ROC) was obtained by using Python 3.5, Matplotlib 3.0.2 and ROC module. A 2-sided *p*-value < 0.05 was considered statistically significant.

## 3. Results

### 3.1. Baseline Characteristics

The baseline clinical characteristics of the patients are shown in Table 1. Recurrence occurred in 218 of the 1618 patients at a rate of 13.5%. The average age of 218 patients with recurrence was 59.6 years old, and the average follow-up time was 8.1 months. There were 142 males (65.1%) and 76 females (34.9%) in the recurrence group. The average age of 1400 patients without recurrence was 58.5 years old, and the average follow-up time was 15.6 months. There were 910 males (65.0%) and 490 females (35.0%) in the no-recurrence group. Univariate analysis indicated that left atrial enlargement (LAE) and early recurrence (ER) may be related to late recurrence. Multivariate analysis further confirmed that LAE (HR = 2.11, 95% CI: 1.52–2.94, *p* < 0.01) and ER (HR = 5.97, 95% CI: 4.25–8.40, *p* < 0.01) were the independent prognostic factors of recurrence.

### 3.2. The Performance of the AI Algorithm

After training and validation, the ROC curves were shown (Figure 3). The AUC of the validation set was 0.88 (95% CI: 0.84–0.91) with sensitivity, specificity, accuracy, precision and F1 score of 78.6%, 96.4%, 87.1%, 96.0% and 0.86, respectively. While the AUC of the testing set was 0.84 (95% CI: 0.78–0.89) with sensitivity, specificity, accuracy, precision and F1 score of 72.3%, 95.0%, 92.0%, 69.1% and 0.71, respectively. 

APPLE, BASE-AF2, CAAP-AF, DR-FLASH and MB-LATER scoring systems were used to predict the risk of recurrence, and the AUCs of each scoring system with different cut-offs were shown (Figure 4A–E). The AUCs of APPLE were 0.51–0.56 (optimal 95% CI: 0.49–0.62, cut-off = 0.5), the AUCs of BASE-AF2 were 0.54–0.64 (optimal 95% CI: 0.58–0.70, cut-off = 0.5), the AUCs of CAAP-AF were 0.50–0.55 (optimal 95% CI: 0.49–0.60, cut-off = 4.5), the AUCs of DR-FLASH were 0.50–0.53 (optimal 95% CI: 0.46–0.59, cut-off = 1.5), and the AUCs of MB-LATER were 0.51–0.63 (optimal 95% CI: 0.57–0.69, cut-off = 1.5). The sensitivity, specificity, accuracy, precision and F1 score of the CNN model and the scoring systems with optimal AUC are shown in Table 2. Compared with each scoring system with optimal AUC, the predictive performance of the CNN model was significantly better (0.84 vs. 0.53–0.64, *p* < 0.01).

The clinical prognostic factors were used to predict the risk of recurrence, and the AUCs were shown (Figure 5). The AUC of left atrial enlargement for predicting recurrence was 0.61 (95% CI: 0.55–0.67) with sensitivity, specificity, accuracy, precision and F1 score of 36.9%, 85.2%, 78.8%, 27.9% and 0.32, respectively. While that of early recurrence was 0.68 (95% CI: 0.62–0.75) with sensitivity, specificity, accuracy, precision and F1 score of 44.6%, 92.4%, 86.0%, 47.5% and 0.46, respectively (Table 3). Compared with the prognostic factors, the predictive performance of the CNN model was also better (0.84 vs. 0.61 and 0.68, *p* < 0.01).

## 4. Discussion

In this study, we found that the AI-enabled ECG algorithm could effectively predict the risk of recurrence in patients with pAF after CA (AUC = 0.84). This is the first known AI algorithm to predict the prognosis of atrial fibrillation by using a 12-lead ECG. Presently, the AI algorithm is almost only used for ECG diagnosis. This study has proved that the AI algorithm is also valuable in predicting prognosis. Although CA is an important treatment for atrial fibrillation, postoperative recurrence of CA is still a problem to be solved. The recurrence rate of paroxysmal AF after CA for 12 months is 10–30%, and persistent AF after CA for 12 months is 25–35% [25]. In this study, recurrence occurred in 218 of 1618 patients at a rate of 13.5%, which was consistent with the results of previous studies. This study provided a cheap, widely used and convenient way to identify the risk of recurrence after CA. It may not only help to identify the population who can benefit from catheter ablation, but also help to select the suitable ablation strategy. For patients at high risk of recurrence, additional ablation beyond PVI may be a reasonable option.

The prognostic models of CA mainly include ALARMEc [26], APPLE [16,17,18], ATLAS [10], BASE-AF2 [19], CAAP-AF [20], DR-FLASH [21], LAGO [27] and MB-LATER [22,23,24]. The sizes of the study cohorts vary greatly among the scoring systems. The parameters included in each scoring system are the independent prognostic factor in the study cohorts. Due to the heterogeneity of the study cohorts, the parameters of the scoring systems are quite different and even in conflict. However, almost all the scores consider left atrial enlargement as a risk factor for recurrence. The previous study indicated that early recurrence might be the most powerful predictor of AF recurrence [28]. Generally, the scores obtained in large study cohorts (ATLAS, APPLE and CAAP-AF) may be more universal than those obtained in small study cohorts (BASE-AF2, ALARMEc and MB-LATER). However, no score has been determined to be superior to others. Although external validation is an important quality standard of risk stratification tools, not every scoring system has been validated. In fact, the ALARMEc, APPLE and MB-LATER scores are the only validated scoring systems [5]. In addition, all scores can only moderately predict the recurrence of atrial fibrillation, and the AUC of each score for predicting recurrence is around 0.6–0.7 [4,5].

The CNN model of this study was trained and validated in a development cohort of 1133 patients and tested in a validation cohort of 485 patients. Compared with the current predictive models, the sample size of this study was large, and the CNN model had better predictive performance than the current predictive models. Since some parameters of ALARMEc, ATLAS and LAGO were not recorded, these scoring systems were not included in the comparison.

Our previous research has proved that AI-enabled ECG algorithms can effectively identify left atrial enlargement [14]. When using 12-lead ECG to predict the risk of recurrence after CA, the CNN model may have taken advantage of the features reflecting left atrial enlargement. Therefore, the inclusion of left atrial enlargement as a predictor may not increase the predictive performance. Although early recurrence has been shown to be a strong predictor of late recurrence, using early recurrence as a predictor cannot predict the risk of late recurrence at baseline. Thus it may limit the model to find out the patients who can benefit from catheter ablation and select the reasonable methods of CA. Therefore, the two independent prognostic factors were not combined with the CNN model to improve the predictive performance.

The predictive performance of the scoring systems was slightly worse than that of previous studies. This may be due to the heterogeneity of the study population. In this study, only left atrial enlargement and early recurrence were independent prognostic factors of recurrence. Therefore, the prediction model with multiple parameters did not benefit from other parameters. Moreover, the type of AF was included as a parameter to predict the risk of recurrence in previous scoring systems. However, in the case of limited samples, the inclusion of different types of AF may lead to excessive dispersion of features and affect the prediction performance of the model. In addition, there may be significant differences in the methods of CA for patients with non-paroxysmal AF, which may also affect the risk of recurrence. Therefore, only patients with paroxysmal AF were included in this study for analysis. In the future, better models can be constructed by expanding the sample size and combining with multi-omics, such as imaging, to increase the predictive range and further improve the predictive performance.

## 5. Limitations

The study has several limitations. As a study about deep learning, the sample size was small. Although deep learning models are also applicable in small datasets [29,30], training and validating with a large sample size can improve the applicability and accuracy. Moreover, in order to further validate the predictive performance, a prospective study is needed. Secondly, this model can only predict the risk of recurrence 3–12 months after CA. However, the ability to predict the risk of recurrence after more than 12 months remains to be verified. One of the key limitations of existing deep learning models is interpretability. The features the neural networks use to predict are unknown. Although the enrolled patients met the diagnostic criteria for pAF, detailed information, such as the number of episodes, duration and use of antiarrhythmic drugs, was not recorded and analyzed in this study. Besides, since all ECG data was collected in an “.XML “ format, images drawn from the data cannot measure parameters just like conventional ECG images. The effect of electrocardiographic parameters such as PR interval and P wave on recurrence was not analyzed.

## 6. Conclusions

In this study, we found that AI can effectively predict the risk of postoperative recurrence in patients with paroxysmal atrial fibrillation by identifying 12-lead ECG characteristics before ablation, and the prediction performance was better than the existing prediction models. However, the results of this study need to be further calibrated and validated using high-quality prospective studies.

## Figures and Tables

**Figure 1 jcm-12-01933-f001:**
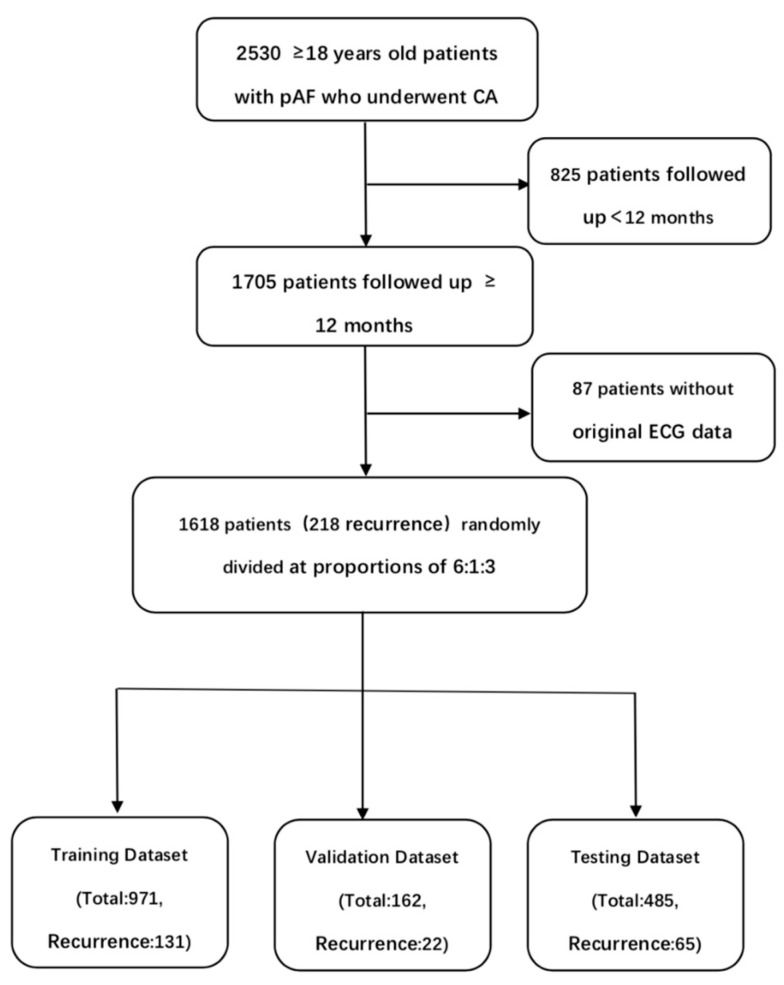
Patient flow diagram.

**Figure 2 jcm-12-01933-f002:**
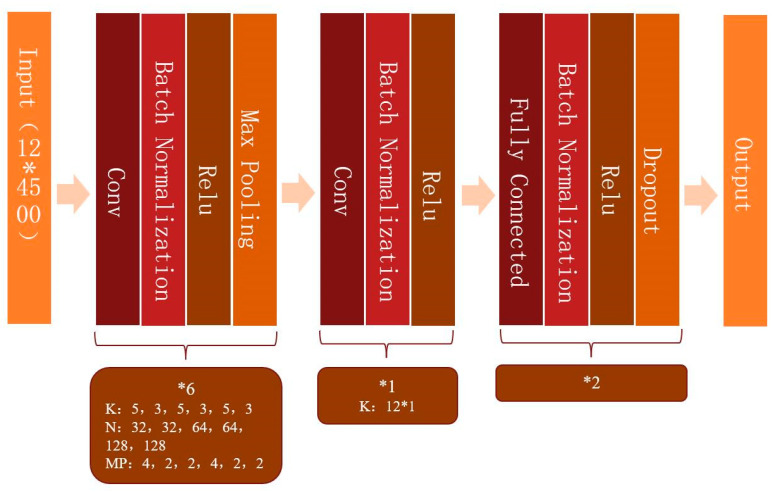
Model diagram.

**Figure 3 jcm-12-01933-f003:**
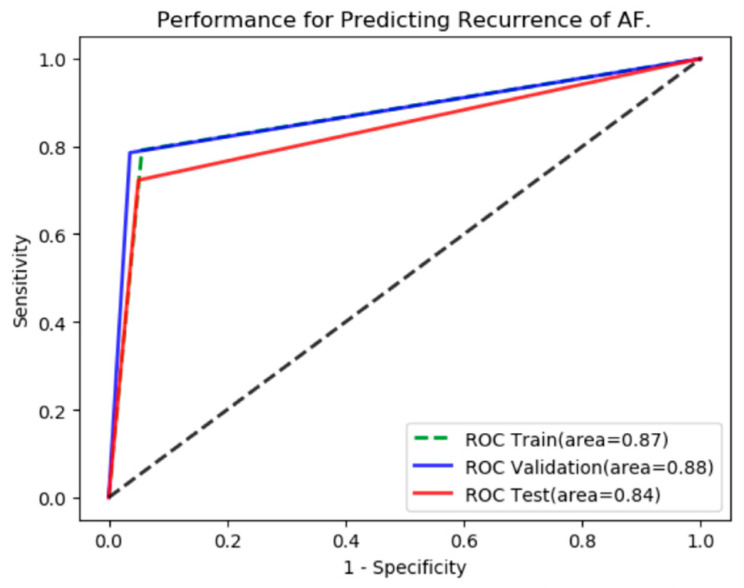
The Predictive Performance of AI Model.

**Figure 4 jcm-12-01933-f004:**
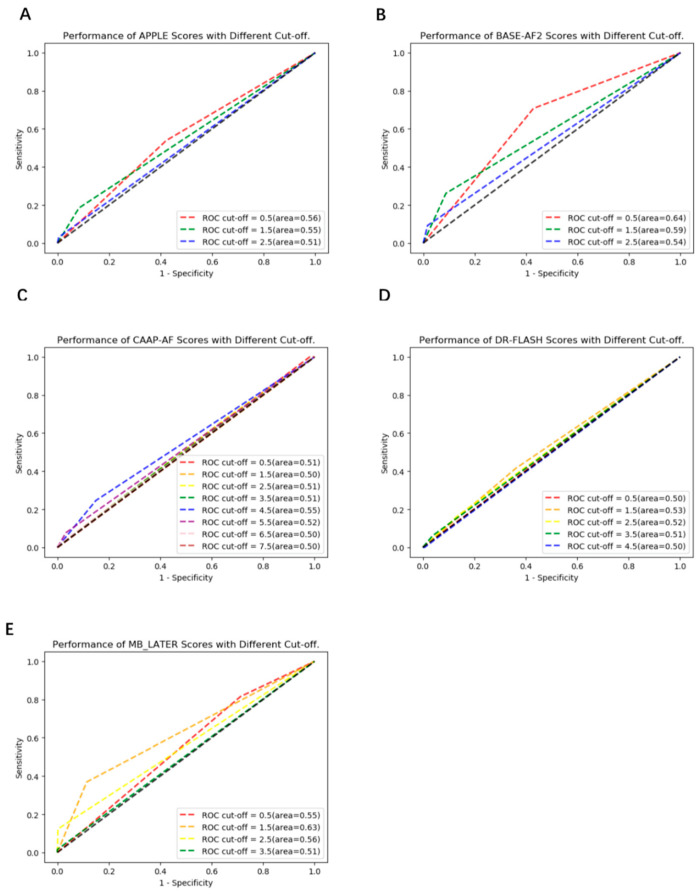
The Predictive Performance of Different Models. (**A**) the AUCs of APPLE, (**B**) the AUCs of BASE-AF2, (**C**) the AUCs of CAAP-AF, (**D**) the AUCs of DR-FLASH, (**E**) the AUCs of MB-LATER.

**Figure 5 jcm-12-01933-f005:**
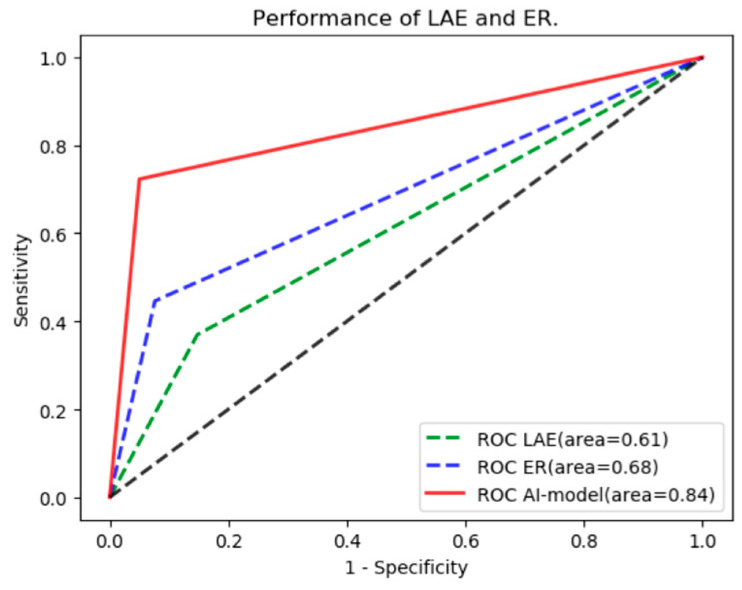
The Predictive Performance of Prognostic Factors.

**Table 1 jcm-12-01933-t001:** Clinical Characteristics of Patients.

		No Recurrence	Recurrence	Univariate Analysis	Multivariate Analysis
Age (Mean) (Year)		59.6	58.5	0.27	——
Gender	Male	910	142	0.97	——
	Female	490	76		
BMI Grade (kg/m^2^)	<18.5	33	8	0.37	——
	18.5–25	1012	146		
	25–30	323	58		
	≥30	32	6		
Smoking	yes	197	27	0.50	——
	no	1203	191		
Alcohol	yes	73	14	0.46	——
	no	1327	204		
Hypertension	yes	517	78	0.74	——
	no	883	140		
Diabetes	yes	140	19	0.55	——
	no	1260	199		
Stroke	yes	106	17	0.91	——
	no	1294	201		
Vascular Disease	yes	93	11	0.37	——
	no	1307	207		
Coronary Artery Disease	yes	127	21	0.79	——
	no	1273	197		
Bundle Branch Block	yes	113	20	0.58	——
	no	1287	198		
Cardiomyopathy	yes	17	4	0.45	——
	no	1383	214		
Left Atrial Enlargement	yes	258	74	<0.01	<0.01
	no	1142	144		
Left Ventricular Enlargement	yes	16	2	0.77	——
	no	1384	216		
LVEF	≥50%	1357	9	0.41	——
	<50%	43	209		
eGFR	≥60	1304	204	0.81	——
	<60	96	14		
AF Duration (years)	≤6	1255	192	0.48	——
	>6	145	26		
Early Recurrence	yes	118	79	<0.01	<0.01
	no	1282	139		

The left atrial anteroposterior diameter >40 mm on echocardiography was diagnosed as left atrial enlargement.

**Table 2 jcm-12-01933-t002:** The Confusion Matrix of Different Score Systems.

		Predicted		Se (%)	Sp (%)	Acc (%)	Pre (%)	F1 Scores	*p* Value
		No R.	R.						
CNN Model	no R.	399	21	72.3	95.0	92.0	69.1	0.71	——
	R.	18	47						
APPLE	no R.	243	177	53.8	57.9	57.3	16.5	0.25	<0.01
	R.	30	35						
BASE-AF2	no R.	240	180	70.8	57.1	59.0	20.4	0.32	<0.01
	R.	19	46						
CAAP-AF	no R.	357	63	24.6	85.0	76.9	20.3	0.22	<0.01
	R.	49	16						
DR-FLASH	no R.	268	152	41.5	63.8	60.8	15.1	0.22	<0.01
	R.	38	27						
MB-LATER	no R.	372	48	36.9	88.6	81.6	33.3	0.35	<0.01
	R.	41	24						

Abbreviations: R. = Recurrence; Se = sensitivity; Sp = specificity; Acc = accuracy; Pre = precision.

**Table 3 jcm-12-01933-t003:** The Confusion Matrix of Prognostic Factors.

		Predicted		Se (%)	Sp (%)	Acc (%)	Pre (%)	F1 Scores	*p* Value
		no R.	R.						
AI Model	No R.	399	21	72.3	95.0	92.0	69.1	0.71	——
	R.	18	47						
LAE	no R.	358	62	36.9	85.2	78.8	27.9	0.32	<0.01
	R.	41	24						
ER	no R.	388	32	44.6	92.4	86.0	47.5	0.46	<0.01
	R.	36	29						

Abbreviations: R. = Recurrence; Se = sensitivity; Sp = specificity; Acc = accuracy; Pre = precision; LAE = left atrial enlargement; ER = early recurrence.

## Data Availability

The data used to support the findings in this study is available from the corresponding author upon reasonable request.

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
