# Peer review of "An Artificial Intelligence-Enabled ECG Algorithm for Predicting the Risk of Recurrence in Patients with Paroxysmal Atrial Fibrillation after Catheter Ablation"

_jcm, 2023, doi:10.3390/jcm12051933_

Round 1
Reviewer 1 Report
This paper is focused on an interesting tool to predict recurrences of atrial fibrillation in patients undergoing CA. The article is well written and the method is clearly presented in a corrected language.
MINOR COMMENTS
Line 34- Recurrence of AF after CA is also dependent on operators and this should be indicated
Line 78/79- “Then the training and validation set were expanded 500 times to produce more 78 data by shifting the start point”. This part should be expressed more clearly.
Line 112- Althought you previously signaled it, in this case it should be used “late recurrence” as well as in line 191
MAJOR COMMENTS
A point which should be discussed is what other electrocardiographic parameters other than signs of LA enlargement could have influenced the sensitivity and specificity of the algorithm (i.e. signs of LV hypertrophy, baseline heart rate, PR interval, RA enlargement). For example, PR interval and RA enlargement have been supposed to be risk factors for AF recurrence after catheter ablation (Zhao L 2013: Why atrial fibrillation recurs in patients who obtained current ablation endpoints with longstanding persistent atrial fibrillation, J Interv Card Electrophysiol Int J Arrhythm Pacing 37(3):283-290; Park J, 2014: Prolonged PR interval predicts clinical recurrence of atrial fibrillation after catheter ablation. J Am Heart Assoc 3(5):e001277). In the table of the clinical characteristics the above mentioned parameters are missing. Are the authors able to retrieve these values? Otherwise They should flag the absence of this data as an important limitation.
Author Response
Thank you so much for your advice.
MINOR COMMENTS
Line 34- Recurrence of AF after CA is also dependent on operators and this should be indicated.
It has been corrected, and the effect of operators has been mentioned.
Line 78/79 - “Then the training and validation set were expanded 500 times to produce more 78 data by shifting the start point”. This part should be expressed more clearly.
The method has been explained in detailed. (Line 81-85)
Line 112- Althought you previously signaled it, in this case it should be used “late recurrence” as well as in line 191
It has been corrected. (Line 123 and Line 206/207)
MAJOR COMMENTS
This study only analyzed the factors included in the previous scoring systems. Considering that the AI algorithm automatically detects all hidden features in the ECG, the influence of other ECG parameters on prognosis was not further analyzed. Since ECG is not an accurate way to diagnose left atrial enlargement, the left atrial anteroposterior diameter >40 mm on echocardiography was diagnosed as left atrial enlargement (LAE) (mentioned in Line 127). As you mentioned, electrocardiographic parameters such as PR interval and RA enlargement have been supposed to be risk factors for AF recurrence after catheter ablation. The predictive performance of the AI algorithm can be further verified by comparing the AI algorithm with the electrocardiographic parameters. However, since all ECG data was collected in an ".XML " format, images drawn from the data cannot measure parameters such as PR intervals just like conventional ECG images. And the absence of this data has been mentioned in the section of limitation (Line 233-236).
Please see the attachment.

Reviewer 2 Report
Summary:
The authors propose a CNN that can predict the risk of AFib recurrence 3-12 months after catheter ablation.
General concept comments:
This is a well written paper that is scientifically sound with appropriate study design and validation.
I suggest the following changes to enhance the quality of the article:
-
Mention “12-lead” ECG in all sections of the paper including the abstract
-
Introduction should provide sufficient related works and must include all relevant references related to AFib recurrence prediction (especially works involving machine learning and deep learning)
-
The authors need to mention the main contributions of the article
-
The author should consider providing model diagram to enhance readability
-
Sub-group analysis results of the testing set across sub-groups defined by age and sex for the model should be provided
-
The scoring systems such as APPLE, BASE-AF2, CAAP-AF, DR-FLASH and MB-LATER should be briefly explained in the Appendix for completeness.
Author Response
Thank you so much for your advice.
- “12-lead” ECG has been mentioned in all sections of the paper including the abstract.
- References related to AF recurrence prediction has been mentioned in the section of introduction. (Line 43-46)
- The main contributions of the article has been mentioned in the section of discussion. (Line 174-178)
- The model diagram has been provided. (Line 100)
- Since there was no statistical difference in gender and age between the recurrence group and the non-recurrence group in this study, and gender and age were one of the parameters in some scoring systems, further subgroup analysis was not carried out to avoid affecting the predictive performance comparison between AI algorithm and other scoring systems.
- Appendix about definitions of different scoring systems has been provided. (Line 361)
Reviewer 3 Report
The study aimed to predict the risk of recurrence in patients with paroxysmal AF (pAF) after CA by an artificial intelligence(AI)-enabled electrocardiography(ECG) algorithm.
1618 ≥18 years old patients with pAF who underwent CA in from January 1, 2012 to May 31, 2019 were enrolled. Compared to several scores of AF- risk, the AI-enabled ECG algorithm seemed to be an effective method to predict the risk of recurrence in patients with pAF after CA.
Major comments
It is known that algorithms are more efficient that certified but not experts cardiologists and for routine and non-complex ECG interpretations. This is a way not to spend time interpreting the ECG.
The manuscript is well-written and detailed. The number of patients is important.
However the manuscript is difficult to read by a general cardiologist (too many abbreviations and complex statistical analysis) and the clinical interest is debatable.
Moreover, simple explanations on AI are required: ref 9 should be clearly summarized in methods. Only in discussion it is indicated that ref 9 has proved that AI-enabled ECG algorithm can effectively identify left atrial enlargement.
Please can you present less statistical tests and more clinical data.
Minor comments
Abstract:
-the AUC ??? (no definition) (repeated several times and only defined p 4)
-F1 score ????
- APPLE, BASE-AF2, CAAP-AF, DR-FLASH and MB-LATER???
In fact there is no description of the artificial intelligence or the criteria used with this method.
Material and methods
-The definition of pAF requiring ablation is missing (number of episodes, duration, number of drugs used for the prevention…). This is an important risk factor of AF recurrence after ablation.
-“In order to train and validate the convolutional neural network (CNN) model, the patients were randomly divided into the training set (n=971), the validation set (n= 162) and the testing set (n=485) at proportions of 6:1:3(Figure 1)” unclear; please explain the significance of each item.
Results
Table 1 reports a lot of data allowing us to conclude that echocardiographic data are the only predictors of the of AF recurrence. Why the simplest sign, the duration of the P wave on the ECG is not reported? (Agarwal YK, Aronow WS, Levy JA, Spodick DH. Association of interatrial block with the development of atrial fibrillation. Am J Cardiol 2003; 91: 882).
Most of the compared studies used different criteria for the prediction of AF recurrence.
Conclusions
“This will help clinicians to develop personalized ablation strategy and postoperative treatment plan”: completely speculative (what is personalized ablation strategy??)
Author Response
Thank you so much for your advice.
Appendix about definitions of different scoring systems has been provided. (Line 361) This may help to read.
More details about the methods and the model diagram has been provided. (Line 81 and Line 100)
According to ESC guideline, considering patient’s choice, catheter ablation is a IIa recommendation. Since number of episodes, duration, number of drugs used for the preventions are parameters in some scoring systems, for avoiding affecting the predictive performance comparison between AI algorithm and other scoring systems, all patients with pAF accepted catheter ablation has been enrolled.
The purpose of each data set has been supplemented. (Line 75-78)
This study only analyzed the factors included in the previous scoring systems. Considering that the AI algorithm automatically detects all hidden features in the ECG, the influence of other ECG parameters on prognosis was not further analyzed. Since ECG is not an accurate way to diagnose left atrial enlargement, the left atrial anteroposterior diameter >40 mm on echocardiography was diagnosed as left atrial enlargement (LAE) (mentioned in Line 127). As you mentioned, electrocardiographic parameters have been supposed to be risk factors for AF recurrence after catheter ablation. The predictive performance of the AI algorithm can be further verified by comparing the AI algorithm with the electrocardiographic parameters. However, since all ECG data was collected in an ".XML " format, images drawn from the data cannot measure parameters such as PR intervals just like conventional ECG images. And the absence of this data has been mentioned in the section of limitation (Line 233-236).
The conclusion has been corrected (Line 244). And the main contributions of the article has been mentioned in the section of discussion. (Line 174-178)
Round 2
Reviewer 1 Report
All the suggestions I gave were satisfied and the definite version is now improved. I do not have additional updates or variations to send to the authors.
Author Response
Thank you very much for your valuable comments and suggestions.
Reviewer 3 Report
The manuscript remains very difficult to read by a general cardiologist and the clinical interest is debatable.
I did not any response to my remarks: Abstract; lines 20; 21 AUC F1 still without definition; no definition of pAF (only ref on ESC recommendations). We have to know the number of pAF, the number of antiarrhythmic prescriptions or at least a reference of the group detailing the study population.
I see only the adjunction of 12 lead ECG repeated several times and without interest.
Line 238: why this sentence? (“The effect of electrocardio-graphic parameters such as PR interval on recurrence was not analyzed”): I do not see the interest of the PR interval study for the evaluation of AF- risk recurrence.

Author Response
Thank you very much for your advice. I'm sorry I didn't give you a satisfactory answer last time.
The definitions of AUC and F1 score have been provided.(Line 19 to 23)
Although the enrolled patients met the diagnostic criteria for paroxysmal atrial fibrillation, detailed information about atrial fibrillation, such as the number of episodes, duration, and use of antiarrhythmic drugs, was not recorded in this study. Since this is an important risk factor of AF recurrence after ablation, the absence of this data has been mentioned in the section of limitation (Line 238-241).
I am sorry that the “12-lead ECG" are suggested to be mentioned in all sections by another reviewer.
Since the PR interval(Park J, 2014: Prolonged PR interval predicts clinical recurrence of atrial fibrillation after catheter ablation. J Am Heart Assoc 3(5):e001277) and P wave were shown to be closely associated with the recurrence of AF after catheter ablation, both of them have been mentioned in the section of limitation. (Line 244)